# Plant SWEET Family of Sugar Transporters: Structure, Evolution and Biological Functions

**DOI:** 10.3390/biom12020205

**Published:** 2022-01-25

**Authors:** Jialei Ji, Limei Yang, Zhiyuan Fang, Yangyong Zhang, Mu Zhuang, Honghao Lv, Yong Wang

**Affiliations:** Key Laboratory of Biology and Genetic Improvement of Horticultural Crops, Ministry of Agriculture, Institute of Vegetables and Flowers, Chinese Academy of Agricultural Sciences, Beijing 100081, China; yanglimei@caas.cn (L.Y.); fangzhiyuan@caas.cn (Z.F.); zhangyangyong@caas.cn (Y.Z.); zhuangmu@caas.cn (M.Z.); lvhonghao@caas.cn (H.L.); wangyong03@caas.cn (Y.W.)

**Keywords:** SWEET, sugar transporter, phloem loading, plant–pathogen interaction, abiotic stress

## Abstract

The SWEET (sugars will eventually be exported transporter) family was identified as a new class of sugar transporters that function as bidirectional uniporters/facilitators and facilitate the diffusion of sugars across cell membranes along a concentration gradient. SWEETs are found widely in plants and play central roles in many biochemical processes, including the phloem loading of sugar for long-distance transport, pollen nutrition, nectar secretion, seed filling, fruit development, plant–pathogen interactions and responses to abiotic stress. This review focuses on advances of the plant SWEETs, including details about their discovery, characteristics of protein structure, evolution and physiological functions. In addition, we discuss the applications of SWEET in plant breeding. This review provides more in-depth and comprehensive information to help elucidate the molecular basis of the function of SWEETs in plants.

## 1. Introduction

Sugar is an important energy source for plants and a necessary carbon source for the synthesis of many of their metabolic intermediates [1,2,3,4]. Since sugar is the main transportable form of energy, it participates in the storage and transportation of nutrients in plants and plays an important role in signal transduction and resistance to stress [5,6,7,8,9]. The ability of plants to store sugar is essential for their adaptation to endogenous or environmental factors and the economic value of crops, and plants rely on the energy provided by sugars to complete their processes of growth, development and reproduction. In addition, the processes of sugar production, transportation and metabolism not only communicate information on metabolic processes, such as protein, lipid and nucleic acid metabolism, but also communicate information on the metabolism of secondary substances. Moreover, the osmotic potential of cells is affected by the decomposition of sugars, which, in turn, affects the opening and closing of stomata and the division of anthers and plays a central role in the metabolism, growth and development of plants [9,10,11,12,13,14,15].

Sugar in plants is primarily synthesized by photosynthesis in leaves during the day or from the degradation of starch at night [16,17,18,19,20]. During the day, the triose phosphate synthesized in the chloroplast matrix is exported to the cytoplasm and used to synthesize sucrose. At night, the starch in the chloroplast is hydrolyzed into maltose and glucose, which are exported to the cytoplasm and continue to synthesize sucrose. Sucrose invertase and sucrose synthase can metabolize sucrose to monosaccharides [21,22]. The sugar synthesized in the source tissues needs to be transported and distributed to the sink tissues to maintain normal plant growth and development. The transportation and distribution of sucrose, glucose and fructose require the participation of sugar transporters [5].

A variety of monosaccharide and sucrose transporters have been identified in plant plasma membranes or vacuolar membranes, including SUTs (sucrose transporters), MSTs (monosaccharide transporters) and SWEETs (sugars will eventually be exported transporters) [1,5,6]. Because the primary product of photosynthesis is sucrose, it is the main carbohydrate that is transported in plants. It is primarily transported and redistributed over long distances through the phloem [23,24,25,26,27]. The loading and unloading of phloem sucrose and the transportation of sucrose across cell membranes and vacuolar membranes require the participation of SUTs [27,28,29]. The transport process that involves sucrose–proton symporters, SUTs and MSTs, requires energy to complete the transmembrane transport of sugars [24,30]. SWEETs are a new type of sugar uniporters that can transport sugars in two directions and promote the diffusion of sugars along a concentration gradient [5,10,31]. Studies have shown that SWEETs participate in important physiological processes of plant growth and development by regulating the transportation, distribution and storage of carbohydrates [32,33]. This review introduces the progress of research on the discovery, structural characteristics, physiological functions and regulation of stress by the SWEET gene family, which is of substantial significance to enhance plant breeding efforts in the future.

## 2. Discovery of SWEET Sugar Transporters

The fluorescent resonance energy transfer (FRET) sensor is a novel fluorescent signal label that is expressed in plants and animals and quantifies the intensity of fluorescence signal so that changes in the concentrations of substrates, such as sugars, amino acids and ions, at the cellular and subcellular levels, can be monitored in living tissues in real time [34,35]. Chen et al. [35] used the glucose FRET sensor to identify a new type of sugar transporter from *Arabidopsis thaliana*, designated SWEET. SWEET proteins use the concentration gradient of intracellular and extracellular sugars to transport them across membranes instead of relying on the proton gradient [6,10]. Thus, the ability of SWEET proteins to transport sugar does not depend on the pH value of the environment. Moreover, SWEET proteins can transport sugar in both directions across the membrane along a concentration gradient driven by solute potential [6,10]. In other words, SWEET proteins can transport sugar from within cells to the extracellular milieu or from extracellularly to intracellularly along the concentration gradient of sugar. While the MSTs and SUTs that have been identified require coupling with H^+^, they use the H^+^ concentration gradient inside and outside of the cell to transport sugar in one direction across the membrane [27,36,37].

It is hypothesized that many important physiological processes of plants, such as phloem loading and nectar secretion, may require the participation of sugar efflux transporters [36]. However, before the discovery of the SWEET gene family, such sugar transporters had not been isolated, resulting in an incomplete understanding of the molecular and physiological mechanisms of these important physiological processes in plants. The discovery of the sugar efflux transporter SWEET plays a pivotal role in understanding the molecular mechanisms of these important physiological processes in plants.

In fact, SWEETs are widespread in prokaryotes, animals and plants. However, compared with plants, there are fewer members of the SWEET gene family in prokaryotes and animals. For example, *Mycoplasma arthritidis*, *Prochlorococcus marinus*, mice (*Mus musculus*), olive baboon (*Papio anubis*) and humans all have only one SWEET gene [38,39]. A total of 17, 21, 29, 105, 52, 27, 18 and 17 SWEET genes have been identified from the vascular plants *A. thaliana*, rice (*Oryza sativa*), eggplant (*Solanum melongena*), wheat (*Triticum aestivum*), soybeans (*Glycine max*), poplar (*Populus trichocarpa*), pears (*Pyrus* spp.) and grapes (*Vitis vinifera*), respectively [35,38,39,40,41,42,43,44]. Different members of the SWEET family from the same plant can transport different sugars. For example, *A. thaliana* AtSWEET2 transports 2-deoxyglucose, *AtSWEET17* transports fructose, and *AtSWEET11, -12* and *-16* can transport sucrose, glucose and fructose [45,46,47,48]. Moreover, different members of the SWEET family of the same species vary in their characteristics of expression in different tissues [6,35,42]. These results suggest that the SWEETs have a variety of important physiological functions in plants.

## 3. Structural Characteristics of the SWEETs

Before the discovery of SWEETs, the MSTs and SUTs that had been identified in plants were members of the major facilitator superfamily (MFS). The N-terminal and C-terminal ends of these proteins are both located on the intracellular side and typically contain 12 α-helical transmembrane domains (TMs). There is a large cytoplasmic loop located on the intracellular side in the middle of the MFS protein, which divides the protein into two domains [26,49,50]. The N-terminal and C-terminal domains each contain six TMs. The topological structures of these two domains are very similar and exist in a pseudo-quadratic axisymmetric manner. The six TMs that form each domain can be split into two groups of three TMs that symmetrically repeat units in an anti-parallel manner [26,49,50]. This unique folding method is designated MFS fold [51].

The plant SWEETs are members of the MtN3/saliva family (PF03083). Its N-terminus and C-terminus are located on the outside and inside of the cytoplasm, respectively. Plant SWEETs generally contain seven TMs (Figure 1a,b). The fourth TM is less conservative and primarily acts as a link. It divides the protein into two MtN3/saliva domains that each contain three TMs that form a “3-1-3” structure [10]. The three TMs of each MtN3/saliva domain are arranged in the form of “TM1-TM3-TM2” to form a triple-helix bundle (THB) (Figure 1a,b). It is apparent that the topological structure of SWEETs differs significantly from those of MSTs and SUTs. This difference could be an important reason why SWEET can transport sugar intracellularly to extracellularly. In addition, the SWEETs of prokaryotes only contain one MtN3/saliva domain composed of three TMs [52]. Therefore, they have been designated SemiSWEETs. It can be hypothesized that one MtN3/saliva domain in prokaryotes underwent replication or horizontal gene transfer, which is defined as the transmission of DNA between different genomes, during the process of evolution, which led to the production of SWEET proteins in eukaryotes that contain two MtN3/saliva domains.

The results of truncation and complementation experiments demonstrated that SWEETs must undergo oligomerization to form homologous or heteromultimers to transport sugars [52]. The most likely scenario is that the eukaryotic SWEETs form dimers, while the prokaryotic SemiSWEETs form tetramers. *A. thaliana* SWEET proteins can form at least eight homopolymers and 47 heteropolymers [52] (Figure 1c). A high-resolution three-dimensional structural analysis of the bacterial SemiSWEET proteins proved that two SemiSWEET protein monomers form a basic translocation pore unit by forming a symmetrical homodimer [53,54,55,56]. In addition, the tryptophan residue from TM2 and the asparagine residue from TM3 are the key sites for the SemiSWEET protein to be able to transport sugars [53,54,55,56]. Rice *OsSWEET2b* was the first eukaryotic SWEET protein for which a three-dimensional structure was resolved [57]. It showed that a single *OsSWEET2b* protein monomer can form a basic translocation pore unit and that TM4 and THB1 are closely linked to constitute the N-terminal region. In addition, THB2 constitutes the C-terminal region. This explains why the truncation of *AtSWEET1* protein into THB1 + TM4 and THB2 and their co-expression enables the transportation of glucose [52]. In contrast, THB1 and TM4 + THB2 that have been truncated and co-expressed cannot transport glucose [52]. Moreover, owing to the inconsistent tightness of TM4 with THB1 and THB2, THB1 and THB2 are structurally asymmetrical, which obviously differs from the symmetrical arrangement of the two THBs in the SemiSWEET homodimer of prokaryotes [57]. Cysteine residues from TM2, asparagine residues from TM3 and TM7 and phenylalanine residues from TM6 are the key sites for *OsSWEET2* to transport glucose [57]. Recently, the crystal structure of *A. thaliana*
*AtSWEET13* with a resolution of 2.8-Å has been obtained. The researchers observed an inward-facing conformation of *AtSWEET13* with the substrate analog 2′-deoxycytidine-5′-monophosphate bound to the central cavity [58]. There are 10 amino acid residues in the *AtSWEET13* protein that play an important role in the recognition and binding of substrates. They are designated Ser20 from TM1 (Ser20^TM1^), Leu23^TM1^, Asn54^TM2^, Trp58^TM2^, Asn76^TM3^, Ser142^TM5^, Met145^TM5^, Asn176^TM6^, Trp180^TM6^ and Asn196^TM7^. Up to now, structural analysis has revealed that SemiSWEET or SWEET proteins have three conformations: outward open conformation, inward open conformation and occluded conformation. These results laid a structural foundation to elucidate the mechanism by which the SWEET protein binds to substrates and transports sugar. Based on this, a rocking-type motion theory was proposed [53]. In OsWEET2b, the proline residues on TM1, TM2, TM5 and TM6 may be the key factors that promote the transition between different conformations.

It is worth noting that there is a database designated dbSWEET (http://bioinfo.iitk.ac.in/bioinfo/dbsweet/Home, accessed on 7 December 2021) that contains more than 2000 SWEET members from prokaryotes and eukaryotes. This database helps researchers to more effectively analyze the biological functions of SWEET through targeted gene editing and simulation experiments.

## 4. Evolution of the SWEET Gene Family

Yuan and Wang [38] used the transporter classification database (TCDB; http://www.tcdb.org, accessed on 12 November 2021) to select SWEET proteins of different species for a phylogenetic analysis that revealed that the SWEET proteins in different species can be divided into three evolutionary clades. The SWEET proteins of monocots and dicots are members of Clade I, the SWEET proteins of metazoans and mammals are members of Clade II, and those of the bacteria and archaea are members of Clade III. Some members of the *Caenorhabditis elegans* MtN3/saliva family in metamorphosis are also included in Clade III. There is only one MtN3/saliva domain composed of three transmembrane helices in all the bacterial proteins in branch III. A phylogenetic analysis indicates that the widespread distribution of the MtN3/saliva SWEET protein may originate from the SemiSWEET protein of prokaryotes. Domain duplication occurred during the evolution of eukaryotes, resulting in a protein with seven transmembrane α-helices that contains two MtN3/saliva domains.

A further phylogenetic analysis of the SWEETs of 16 types of angiosperms revealed that the family was divided into four subfamilies that are designated Clade I~IV (Figure 2), which is consistent with the results of previous studies [39]. Utilizing *A. thaliana* as an example, the Clade I subfamily contains the three members *AtSWEET1, -2* and *-3*; the Clade II subfamily has five members, including *AtSWEET4, -5, -6, -7* and *-8*; the Clade III subfamily has seven members, including *AtSWEET9, -10, -11, -12, -13, -14* and *-15*; and the Clade IV subfamily has the two members *AtSWEET16* and *-17*. In addition, different subfamilies have selective preferences for monosaccharides or disaccharides. The Clade I and II subfamilies specifically transport hexose, the Clade III subfamily specifically transports sucrose, and the SWEET protein of the Clade IV subfamily is located on the vacuolar membrane and tends to transport fructose [59].

According to the existing evolutionary analysis, it is believed that the SWEET protein containing 7-TMs and having two MtN3/saliva domains in eukaryotes is produced by duplication of the MtN3/saliva domain containing 3-TMs in prokaryotes. This allows for more complex sucrose transport in eukaryotes [38,39,52]. More interestingly, the SWEET proteins of unicellular algae contain 7-TMs but have not yet formed a conserved THB unit, further speculating that multicellular plants (bryophytes and flowering plants) may acquire 3-TMs from symbiotic bacteria through horizontal gene transfer or possibly obtain 3-TMs through internal duplication.

## 5. SWEETs Are Involved in Plant Growth and Development

### 5.1. Participation as a Sucrose Transporter in Phloem Loading

After the photosynthetic products are synthesized in leaves, they must undergo phloem loading, long-distance transportation and phloem unloading in the sink organs to ensure that the photosynthetic products are transported and distributed between the source and sink tissues. Plants that use sucrose as the main transport form of photosynthetic products primarily conduct their phloem loading through the apoplastic pathway. This pathway involves the transmembrane transport of sucrose and must be completed with the assistance of the corresponding sucrose transporter [60]. Before the discovery of SWEETs, it was not clear what type of transporter assisted the transport of sucrose from the phloem parenchyma cells to the apoplast near the sieve-element–companion-cell complex. This is a prerequisite for sucrose to complete phloem loading in the apoplast pathway. Chen et al. [6] found that *AtSWEET11* and *AtSWEET12* that are located on the phloem parenchyma plasma membrane are responsible for this process. This was the first proof that the SWEET serves as a carbohydrate efflux carrier to play a key role in the export of sucrose from the phloem apoplast. It provides the prerequisite preparation for the H^+^/sucrose cotransporter SUT1/SUC2 to transport sucrose from the apoplast to the sieve-element–companion-cell complex, thus, revealing the entire process of phloem apoplast loading (Figure 3).

Rice *OsSWEET11* is a homologous gene of *A. thaliana*
*AtSWEET11* and *AtSWEET12* and is also a carrier of low-affinity sucrose transport that is located on the cytoplasmic membrane and primarily expressed in the phloem of rice leaves [6,61]. Therefore, it is hypothesized that *OsSWEET11* also plays an important role in phloem loading. MtSWEET11 is a nodule-specific sucrose transporter in *Medicago truncatula*, which participates in the distribution of sucrose in nodules [62].

### 5.2. Participation in Male Reproductive Development

The SWEET gene family in plants is usually involved in the development of pollen and is related to plant fertility. Petunia *NEC1* (*AtSWEET9*) is expressed in nectaries and stamens, particularly in anther stomium cells and upper filaments. Inhibition of the expression of *NEC1* leads to premature anther dehiscence, while the pollen is not yet mature, eventually leading to male sterility [63,64]. In *A. thaliana*, *AtSWEET1*, *AtSWEET5*, *AtSWEET7*, *AtSWEET8/RPG1* and *AtSWEET13/RPG2* are all expressed in pollen development and may be involved in reproductive development. *AtSWEET8* is highly expressed in microspore mother cells and the tapetum. The microspore plasma membrane of the *atsweet8* mutant could not form a regular wavy structure during the tetrad period, which caused the sporopollenin to be deposited abnormally and, finally, led to the degradation of pollen [65,66]. *AtSWEET13* is expressed in anthers. Although the pollen exine of the *atsweet13* mutant is slightly defective, the fertility of the mutant does not differ significantly from that of the wild type [66]. *AtSWEET13* can partially compensate to improve the fertility of *atsweet8* mutants. The fertility of *atsweet8**;atsweet13* double mutant was significantly lower than that of the *atsweet8* single mutant [66]. These results indicate that *AtSWEET8* and *AtSWEET13* are partially redundant in terms of plant fertility. *AtSWEET8* primarily affects the fertility of early inflorescence, while *AtSWEET13* primarily affects the fertility of later inflorescence. In rice, *OsSWEET11*, which is located on the plasma membrane, is highly expressed in spikelets and pollen. Pollen development of *OsSWEET11*-silenced plants was arrested at the mononucleate or binucleate stages; the starch content of the pollen is reduced, and the pollen develops poorly, resulting in sterile or semi-sterile plants [67,68,69].

In addition to the above genes, many SWEETs that may be related to male reproductive development have been identified. There are five members, *Cs7g02970*, *Cs3g14550*, *Cs3g20720*, *Cs9g04180* and *Cs2g28270*, of the SWEET gene family in sweet orange, which are expressed abundantly in flowers [70]. Tomato (*Solanum lycopersicum*) *SlSWEET5b/LeSTD1* is the most highly expressed in flowers and specifically expressed in mature pollen grains [41,71]. Seven SWEET gene family members in grapes, including *VvSWEET3*, -*4*, -*5a*, -*5b*, -*7*, -*10* and -*11*, are highly expressed in flowers [40]. More than 20 GmSWEET genes in soybean are highly expressed in flowers [39]. Ten CsSWEETs, including *CsSWEET5a*, -*3a*, -*9a*, -*7b*, -*17a*, -*9b*, -*15c*, -*10c*, -*1b* and -*5c*, were highly expressed in one or several pollen stages of tea (*Camellia sinensis*) [72]. Most SWEET genes of watermelon showed high expression levels in male flowers [73]. During the petunia flower development, the expression level of *PaSWEET13c*, -*9a*, -*1d*, -*5a* and -*14a* increases with the maturation of the flower [74]. During the anthesis, *JsSWEET1*, -*2*, -*5*, -*9*, -*10*, -*16* and -*17* were highly expressed in the flowers of Jasmine (*Jasminum sambac*) [75]. The boron-dependent glucose transporter, *PwSWEET1*, plays a necessary role in *Picea wilsonii* pollen germination and pollen tube growth [76].

### 5.3. Participation in Seed Development

The ability to transport soluble sugars to developing seeds affects the size and weight of seeds, thereby determining the yield of crops, such as corn (*Zea mays*), rice and wheat (*Triticum aestivum*) [77]. The size of seeds increased during the long-term domestication of crops. This selection must be related to the metabolism and transportation of soluble sugars. The expression of the *ZmSWEET4c* gene in maize and its ancestor species teosinte (*Z. mays* L. ssp. *parviglumis*) is quite different [42]. It is speculated that *ZmSWEET4c* may be a locus related to the regulation of sugar transport during maize domestication. Further research found that *ZmSWEET4c* is located on the cell membrane of the basal endosperm transfer layer. The endosperm of the T-DNA insertion mutant of this gene becomes smaller, and the starch content and weight of the seed are significantly lower than those of the wild-type control, displaying the “empty seed coat” phenotype [42]. These results confirm that *ZmSWEET4c* has a key role in the filling of corn seeds. Rice *OsSWEET4* is homologous to *ZmSWEET4c*. The *OsSWEET4* gene expression level, substrate transport and other characteristics, as well as the mutant plant phenotype, are similar to those of maize *ZmSWEET4c*. It is hypothesized that this gene can regulate the filling process of rice seeds and is a site that is related to the regulation of sugar transport that evolved during the process of rice domestication.

During the development of *A. thaliana* seeds, the sucrose efflux vectors *AtSWEET11*, *AtSWEET12* and *AtSWEET15/SAG29* located on the plasma membrane displayed specific temporal and spatial expression in seeds. The embryos of the three mutant *atsweet11;12;15* seeds develop slowly. The seed weight and starch and oil contents are significantly lower than those of the wild type, and the seeds are shrunk and shriveled [78]. In rice, *OsSWEET11*, *OsSWEET14* and *OsSWEET15* are highly expressed in caryopses [79,80,81]. The *ossweet11* mutant exhibited a defect in endosperm development, resulting in an empty-seed phenotype with more starch that accumulated in the pericarp [79]. Moreover, *ossweet14;ossweet11* and *ossweet15;ossweet11* double mutants had a much more severe phenotype than *ossweet11* single mutant, including a strongly reduced grain yield and grain-filling rate and increased starch accumulation in the pericarp [80,81]. These results show that *OsSWEET11*, *OsSWEET14* and *OsSWEET15* play important roles in rice seed filling.

Soybean seeds require a substantial amount of nutrients from the endosperm to maintain their growth during the early stages of development. Insufficient nutrients can lead to seed abortion and a decrease in yield. *GmSWEET15a* and *GmSWEET15b* are primarily expressed in the endosperm of soybean at the cotyledon stage. After these two genes are knocked out, the development of embryos slows, and the content of sugar in the embryo is substantially reduced, which eventually leads to seed abortion [82]. These results indicate that GmSWEET15 plays a role in the development of soybean embryos by mediating the transfer of sucrose from the endosperm to the embryo during early seed development.

### 5.4. Participation in Fruit Development

The soluble sugar content, which primarily refers to sucrose, glucose and fructose, is an important indicator that determines the quality of fruit. Since the SWEET gene family in plants transports sugars, it is logical that they may play a key role in fruit development. The orange (*Citrus sinensis*) genome contains 16 SWEET genes, with *Cs2g28300*, *Cs3g14550*, *Cs7g02970*, *Cs3g14500*, *Cs3g20720*, *Cs2g04140* and *orange1.1t02627* highly expressed in fruit [70]. The levels of expression of six SWEET genes (*VvSWEET4*, -*7*, -*10*, -*11*, -*15* and -*17d*) in grapes increase with the development of berries [40]. Apple (*Malus domestica*) *MdSWEET1.1/2*, *MdSWEET2.4* and *MdSWEET3.5* are expressed at higher levels in young fruit, while *MdSWEET3.6/7* is expressed more abundantly in large fruit [70]. *AnmSWEET5* and *AnmSWEET11* of pineapple (*Ananas comosus*) have high expression levels during the early stage of fruit development [83]. Nine SWEET genes in apples are expressed at high levels during fruit development. Among them, *MdSWEET9b* and *MdSWEET15a* may be involved in the accumulation of sugar in apples [84]. *SlSWEET1b*, *SlSWEET1c*, *SlSWEET2a*, *SlSWEET7a* and *SlSWEET14* in tomato are more highly expressed in young fruit, and the expression levels of these five genes gradually decrease as the fruit mature [41]. *SlSWEET7a* and *SlSWEET14*, which encode membrane localization proteins, are primarily expressed in pedicels, vascular bundles and seeds. The sugar content of tomato fruit in which *SlSWEET7a* or *SlSWEET14* have been silenced increased significantly, and the size of fruit also increased [85]. Artificial domestication selection of these genes is expected to breed high-quality varieties. These data indicate that SWEET may be involved in the transportation and distribution of soluble sugar in fruit, which has a significant impact on their yield and quality.

### 5.5. Participation in Nectar Secretion

Nectaries can secrete nectar, which can lure insects to collect nectar to complete the pollination process, thereby ensuring that plants obtain heterologous genes and ensuring population reproduction and evolution. Although the function and composition of nectar have been clearly understood, the mechanism of nectar secretion was unclear before the discovery of the SWEET gene family. The *NEC1* (a homolog of *AtSWEET9*) gene is primarily expressed in the nectary parenchyma cells in *Petunia hybrida*, and its expression positively correlates with the amount of nectar secreted [63]. Silencing the expression of *NEC1* leads to male sterility [64], but to our knowledge, such studies on the nectar phenotype have not been conducted. Lin et al. [86] found that *A. thaliana AtSWEET9* is located in the plasma membrane and specifically expressed in nectary parenchyma cells. The amount of nectar secreted by the *atsweet9* mutant is reduced, which clearly shows that *AtSWEET9* plays an important role in the secretion of nectar from nectaries [86]. Homologs of *AtSWEET9* have been identified in turnip (*Brassica rapa*) and coyote tobacco (*N. attenuata*). Suppressing the expression of *BrSWEET9* or *NaSWEET9* also resulted in a decrease in the secretion of nectar by mutant plants, indicating that these two genes may also play a key role in the secretion of nectar [86].

### 5.6. Participation in Leaf Senescence

The SWEET gene family in plants is also involved in the regulation of the process of senescence. Rice *OsSWEET5* can transport galactose and is expressed in senescent leaves. The overexpression of this gene causes significant changes in the contents of soluble sugar and indole acetic acid (IAA) in plant leaves, leading to delayed growth and premature senescence in the seedling stage, indicating that the mediation of transport of galactose by *OsSWEET5* plays an important role in plants [87]. However, the *OsSWEET5* gene knockouts did not exhibit any phenotypic changes [87]. This finding indicates that there may be other SWEET genes that can transport galactose that are redundant with *OsSWEET5*. The level of expression of *AtSWEET15* in *A. thaliana* is similar to that of *OsSWEET5* and gradually increases during leaf senescence. Plants that overexpressed *AtSWEET15* exhibited a phenotype of delayed growth and accelerated leaf senescence, but the T-DNA insertion mutant of *AtSWEET15* did not differ significantly from that of the wild-type plants [88,89]. *PbSWEET4* of pear (*Pyrus bretschneideri*), which is homologous to *AtSWEET15* of *A. thaliana*, has been shown to regulate senescence and the content of leaf sugar and is highly expressed in old leaves. The overexpression of *PbSWEET4* in strawberries (*Fragaria* × *ananassa*) can reduce the contents of sugar and chlorophyll in leaves and accelerate the senescence of leaves [90]. These studies on the overexpression of certain SWEET genes suggest that this change may disrupt the correct distribution and flow of soluble sugar or cause soluble sugar extravasation, which will negatively affect plant growth.

## 6. SWEETs Participate in the Interaction between Host Plants and Pathogens

The SWEET gene family plays a key role in plant–pathogen interactions [91,92,93,94,95]. When bacterial or fungal pathogens invade plants, they secrete virulence proteins, described as transcription activator-like (TAL) effectors that can bind to the promoters of specific SWEET genes and activate their expression, which helps the pathogens to obtain sugars from plants for their growth and reproduction (Figure 4). Rice *OsSWEET11-15* has been proven to provide nutrition for rice bacterial blight (*Xanthomonas oryzae* pv. *oryzae*) [68,93,95,96,97,98,99,100,101]. *X*. pv. *oryzae* infects rice and secretes specific TAL effectors, which combine with the specific elements (effector-binding elements, EBEs) of the promoter of the target *OsSWEET* gene to induce the up-regulated expression of the gene. This leads to the outflow of more carbohydrates so that the pathogenic bacteria can obtain nutrition and multiply. This increases the susceptibility of the plant to disease. TAL effectors that can induce the expression of *OsSWEET11*-*15* have been identified. Among them, four TAL effectors have been identified that can induce the expression of *OsSWEET14* [93,95,96,97,98]. After suppressing *OsSWEET11*, the rice plants were resistant to sheath blight disease caused by the fungus *Rhizoctonia solani* [102].

The phenomenon of pathogen-induced SWEET gene expression is also found in other plant species. Cassava (*Manihot esculenta*) *MeSWEET10a* can be induced by the TAL20 effector of *X**. axonopodis* pv. *Manihotis* [103]. Orange *CsSWEET1* can be induced by the PthA4 and PthAw effectors of bacterial citrus canker (*X. citri* ssp. *citri*) [104]. Grape VvSWEET4 and VvSWEET7 can interact with gray mold (*Botrytis cinerea*) [40,105]. The level of expression of *VvSWEET4* and *VvSWEET7* was up-regulated after *B. cinerea* infected grapes [35,93]. The *atsweet4* mutant is not sensitive to *B. cinerea* [40]. The level of expression of *AtSWEET2* increases sharply after the oomycete *Pythium irregulare* infects the roots of *A. thaliana*, while the *atsweet2* mutant is not sensitive to *Pythium* [45]. This suggests that the location of *AtSWEET2* in the vacuole membrane enables it to provide glucose that *Pythium* can use to grow and reproduce. The accumulation of sugar in the leaves of *A. thaliana sweet11;sweet12* double mutants can trigger a defense pathway against the fungal pathogen *Colletotrichum higginsianum* mediated by salicylic acid [106]. The *AtSWEET* genes in *A. thaliana* are also induced by bacterial speck disease (*Pseudomonas syringae* pv. *tomato* DC3000), powdery mildew (*Golovinomyces cichoracearum*) and clubroot (*Plasmodiophora brassicae*) [35,107]. The level of expression of the SWEET gene (UPA16) in pepper (*Capsicum annuum*) is induced by black rot (*X. campestris* pv. *vesicatoria*) [108]. The level of expression of the SWEET gene in wheat is induced by the rust fungus *Puccinia striiformis* [109]. GhSWEET10 in cotton (*Gossypium hirsutus*) can be induced by Avrb6, a TAL effector that determines the pathogenicity of *X. citri* subsp. *malvacearum* (*Xcm*) [110]. Silencing *GhSWEET10* reduces the susceptibility of cotton to *Xcm* [110]. The infection of *P. brassicae* on *B. rapa* can induce the translocation of sugar between the tissues that produce sugar and that of the susceptible clubbed tissue. BrSWEETs participate in this process [111]. Unlike the above cases, *IbSWEET10* in sweet potato (*Ipomoea batatas*) plays a positive role in resistance to the fungus *Fusarium oxysporum*. The overexpression of *IbSWEET10* can increase the resistance of sweet potato to *F. oxysporum* by reducing the sugar content of sweet potato [112].

The results described above indicate that the SWEET gene family participates in host–pathogen interactions, but only a few SWEET genes and mechanisms of their interaction with pathogens have been explained. Elucidating these mechanisms will help researchers to modify the binding sites of pathogens through molecular biological methods to obtain crops that are resistant to pathogens. A number of bioinformatics tools, such as PrediTALE, Storyteller, Talvez, TALgetter and Target Finder, can be used to identify and predict EBEs for almost all known TAL effectors [113]. Resistant plants can be obtained by modifying EBEs using gene-editing technology. For example, The CRISPR-Cas9 system was used to edit the EBEs in the promoter regions of *SWEET11*, *SWEET13* and *SWEET14*, resulting in rice plants that were resistant to the main pathogens that cause rice blight [114]. In this regard, we systematically summarize a strategy to obtain disease-resistant plants by modifying SWEET genes (Figure 5). Moreover, some studies have shown that SWEET family genes not only provide carbohydrates for pathogenic bacteria but also provide nutrients for beneficial microorganisms. For example, alfalfa *MtSWEET11* can be induced by rhizobia to provide carbohydrates for the formation of symbiotic nodules [62].

## 7. SWEETs Are Involved in Plant Responses to Stress

Soluble sugar is an important source of energy and material in cells and plays an important role in the regulation of stress responses. Plants that are subjected to stress maintain the balance of cell osmotic potential by regulating the redistribution of soluble sugars in the tissues to help the plants maintain normal growth [115]. Sugar transporters are key factors that regulate the redistribution of soluble sugars, which can respond to a variety of stresses. During the natural cold acclimation process of tea tree (*Camellia sinensis*), the levels of expression of *CsSWEET2*, *CsSWEET3* and *CsSWEET16* were significantly suppressed, while the levels of expression of *CsSWEET1* and *CsSWEET17* increased sharply [116]. CsSWEET16, which is located in the vacuole membrane, can regulate the tolerance of plants to cold. The overexpression of *CsSWEET16* in *A. thaliana* resulted in plants that were less damaged by cold stress [117]. Several cis-acting elements related to stress and hormone responses were identified in the upstream promoter region of the tomato *SlSWEET* gene. The expression of multiple *SlSWEET* genes in the leaves, roots and fruit at the green and red maturity stage changed significantly when the plants were subjected to high sugar, high salt and high/low temperatures [41]. An expression analysis showed that the banana (*Musa* spp.) MaSWEETs could be involved in cold, salt and osmotic stress [118]. Cabbage BoSWEETs have been found to be involved in the improvement of tolerance to cold stress [119].

The response of AtSWEET genes to abiotic stress in *A. thaliana* has been studied in-depth. *AtSWEET16* and *AtSWEET17* are homologs, and both are involved in the responses to abiotic stress. The content of fructose in the leaves of the *atsweet17* mutant plants increased significantly when the plants were subjected to stress from low nitrogen and cold, and the growth of roots was significantly reduced [46,120]. Under cold stress conditions, the content of fructose in the leaves of plants that overexpressed *AtSWEET17* decreased by 80%, but the root growth increased significantly [46,120]. These results indicate that *AtSWEET17* is responsible for the two-way transport of fructose to maintain the balance of fructose in the cytoplasm of *A. thaliana* leaves and roots to improve the plant’s tolerance to abiotic stresses, such as low nitrogen and cold stress. Chilling injury, osmotic stress and low nitrogen can all cause a decrease in the level of expression of *AtSWEET16* [47]. The overexpression of *AtSWEET16* caused the soluble sugar content in the plant to differ substantially from that of the wild-type plant, and the rate of seed germination and tolerance to cold increased [47]. The efficiency of growth and nitrogen utilization in plants that overexpressed *AtSWEET16* was higher than that of the wild type when the plants had a sufficient supply of nitrogen. However, the wild type could more effectively utilize nitrogen compared with the overexpression plants subjected to low-nitrogen stress [47]. These results indicate that *AtSWEET16* is involved in a variety of abiotic stresses, and its function in these abiotic stresses may be relatively independent. The expression level of *AtSWEET15* of *A. thaliana* gradually increased during the natural senescence of leaves. Low temperature, drought and high-salt stress can all induce the expression of *AtSWEET15*. The induced expression under this osmotic stress depends on the abscisic acid pathway [88,89,121]. The tolerance of *atsweet15* mutant plants to high-salt stress was significantly higher than that of the wild-type plants [89]. The overexpression of *AtSWEET15* not only accelerated the senescence of plant leaves but also increased their sensitivity to salt stress [89]. This could be owing to the decrease in root cell viability of the transgenic plants caused by the overexpression of *AtSWEET15*.

*AtSWEET11* and *AtSWEET12* do not only transport sucrose in the leaves; they also exist in the xylem vessels of flower stems, and they can transport sucrose, glucose and fructose [48]. The stem diameter, stem phloem and xylem area of the *atsweet11;12* double mutant plants were significantly reduced under low-temperature stress [48]. However, the low tolerance of double mutant plants to temperature improved significantly [48]. These results indicate that *AtSWEET11* and *AtSWEET12* can transport sugar to the secondary xylem to meet the nutrients required for the formation of secondary cell walls, thereby regulating the tolerance of *A. thaliana* to low-temperature stress [48]. The levels of expression of *AtSWEET11* and *AtSWEET12* are also regulated by water stress [122]. The levels of expression of the three genes *AtSWEET11*, *AtSWEET12* and *AtSUC2* that are involved in sucrose phloem loading in the leaves of *A. thaliana* increased under water deficit conditions, and the ability to transport sucrose from the leaves to the roots increased [122]. Simultaneously, the levels of expression of *AtSUC2* and *AtSWEET11*-*15* in the roots increased [122]. This reveals that these genes may function to unload sucrose from the phloem in roots. These results indicate that plants regulate the redistribution of carbohydrates in their tissues by regulating the expression of sucrose carrier protein under water-deficit stress. In other words, more carbohydrates that are synthesized in the leaves are distributed to the root system, thereby reducing the adverse effect of a water deficit on plants.

## 8. The Role of SWEETs in Plant Ion Transport

The SWEET gene family in plants also plays an important role in ion transport. *A. thaliana* roots were treated with 25 μmol/L aluminum ion [123]. This resulted in the up-regulation of the expression of *AtSWEET13* by nearly 160-fold, indicating that this gene may play an important role in maintaining the content of aluminum in the roots [123]. After soybean seedlings were treated with iron deficiency for 1 h, the level of expression of *Glyma05g38351* (the homologous gene of *AtSWEET12*) in the leaves increased by approximately 3-fold, while the expression level of the *Glyma05g38340* and *Glyma08g01310* genes that are homologous to *AtSWEET13* decreased, indicating that these three SWEET genes may be involved in the transport and distribution of iron in soybeans [124]. Barley (*Hordeum vulgare*) seedlings were treated with ammonium nitrogen (NH_4_^+^) or nitrate nitrogen (NO_3_^-^). The level of expression of a SWEET gene that is homologous to *AtSWEET11* in the plants treated with ammonium nitrogen was twice that of the plants treated with nitrate nitrogen [125]. This indicates that the gene may play an important role in regulating the transport of nitrogen between cells. During the formation of a symbiotic nodule of the legume *M. truncatula*, boron deficiency will lead to a decrease in the level of expression of the SWEET gene family. However, when calcium ions are provided to the plant, the expression levels of these down-regulated SWEET genes rapidly recover, indicating that the SWEET gene family may play an important role in the balance of calcium/boron ions [126]. 

The function of SWEETs in the transport of copper ions has been thoroughly studied. Yuan et al. [69] used the conserved domain of the rice *OsSWEET11* as bait to screen the two copper transporters COPT1 and COPT5 located on the plasma membrane using a yeast two-hybrid system. The yeast mutant strain MPY17 has a deletion in the copper ionophore function. Only the simultaneous expression of *OsSWEET11*, *COPT1* and *COPT5* can restore the function of yeast to transport copper [69]. This indicates that the three proteins interact to form a copper ion transporter complex on the cell membrane, which performs its function of transporting extracellular copper ions into the cell. The overexpression of *OsSWEET11*, *COPT1* and *COPT5* in rice resulted in an increase in the content of copper ions in the aboveground tissues and roots but a decrease in the content of copper ions in the xylem sap [69]. This indicates that *OsSWEET11* can affect the redistribution of copper ions in rice and regulate the transport of copper ions.

## 9. Conclusions and Future Perspectives

The transportation and distribution of sugar play important roles in regulating plant growth and development and responding to biotic and abiotic stresses. Therefore, it is particularly necessary to study how sugar transporters transport and redistribute sugar in various stages of plant growth. In the past ten years, there have been many important advances in the research of SWEET proteins, but there are still many problems that have not been resolved. The function of the plant SWEET gene family is closely related to its structure. However, the three-dimensional structures of only two plant SWEET proteins have been resolved to date, and only one conformation that opens into the cell has been observed. The three conformations of the same SWEET protein have not been observed in the same species. These results are not enough to fully understand the mechanism that SWEET proteins use to transport sugar. Therefore, further strengthening the crystal structure analysis of SWEET proteins with different sugar transport functions will help to clarify how multiple SWEETs in plants distinguish and recognize their respective substrates and the key factors that promote the conversion between different conformations. Most plants contain multiple members of the SWEET gene family. How these genes work together, how they are regulated, whether they are regulated at the transcriptional or translational level and how they achieve functional diversification still remain unclear. In addition, although SWEET genes have been found in most plants, the functional research on the SWEET gene is primarily focused on *A. thaliana* and rice, while the functional research on other crops needs to be strengthened to fully reveal the diversified functions of SWEET genes.

Research on SWEET sugar transporters in the past decade has shown that SWEETs are involved in important physiological processes, such as plant phloem loading, pollen development, nectar secretion, seed filling, leaf senescence and fruit development. They also play key roles in host–pathogen interactions and various responses to abiotic stress. The use of molecular methods and technologies to regulate the expression of SWEET genes and artificially control the flow of carbohydrates has a substantial potential to improve crop yields, quality and breed varieties that are resistant to disease and stress. Such research will be of substantial value to enable stakeholders to respond to the global food crisis.

## Figures and Tables

**Figure 1 biomolecules-12-00205-f001:**
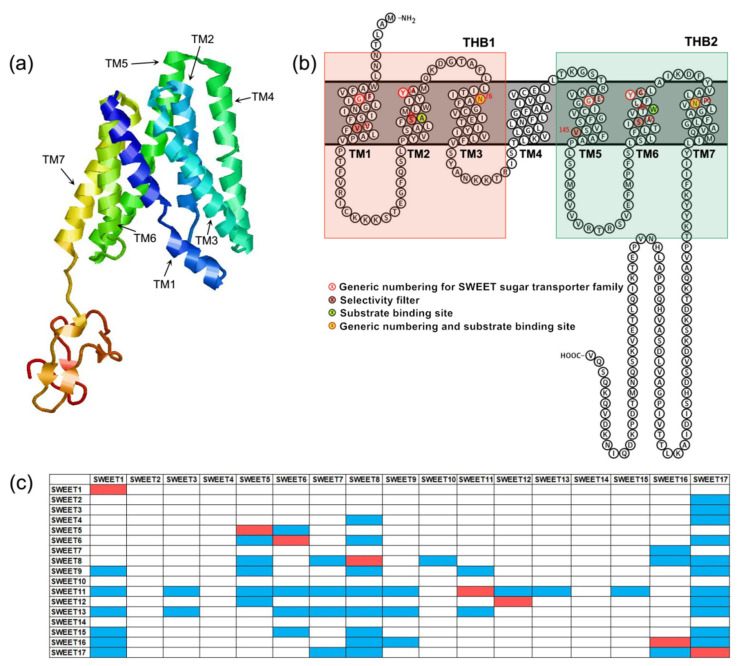
Structural characteristics of SWEETs. (**a**) The three-dimensional model of AtSWEET13 constructed with Phyre2. (**b**) Snake diagram of AtSWEET13 with key positions labeled. (**c**) AtSWEETs interaction matrix diagram [52]. The colored boxes represent interaction between two SWEETs. The red boxes indicate homopolymers, and the blue boxes indicate heteropolymers.

**Figure 2 biomolecules-12-00205-f002:**
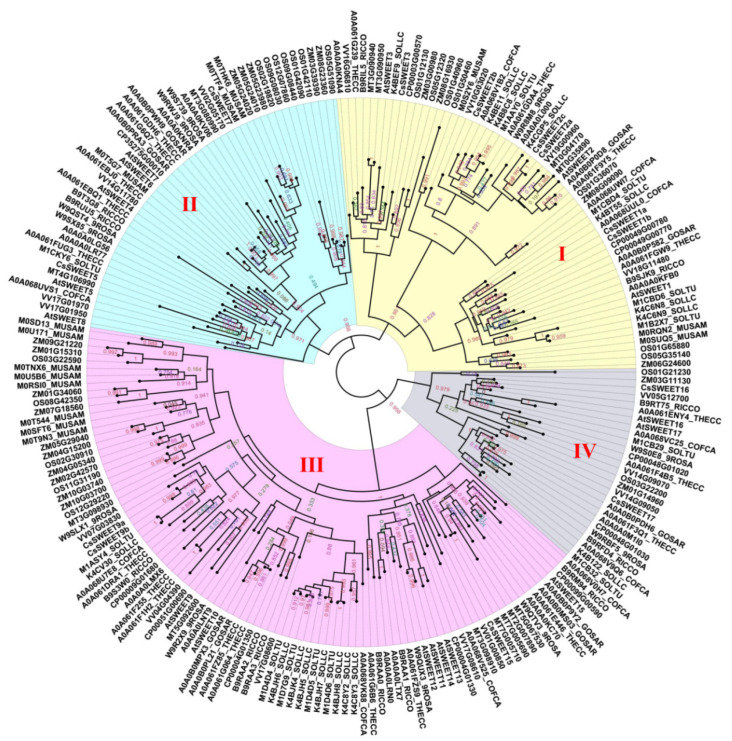
A phylogenetic tree of the SWEETs of 16 types of angiosperms. Accessions were identified from dbSWEET and UniProt. An alignment was conducted using MAFFT (v7.037). Phylogenetic trees were constructed using FastTree (http://www.microbesonline.org/fasttree/, accessed on 25 November 2021) based on the JTT + CAT model. The numerical values in the figure represent the reliability of the clades. The closer the value of the clade is to 1.0, the higher the confidence.

**Figure 3 biomolecules-12-00205-f003:**
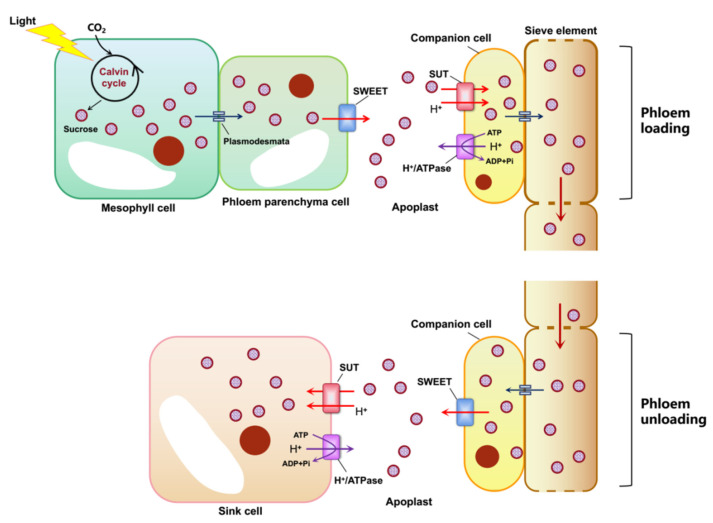
Diagram of phloem loading and unloading. Sucrose synthesized by photosynthesis in mesophyll cells is transported to phloem parenchyma cells through plasmodesmata and then transported to apoplasts near the sieve-element–companion-cell complex (SE-CC) through the SWEET protein. The sucrose is transported to the SE-CC through the H^+^/sucrose cotransporter SUT1/SUC2 and transported to the sink tissue over a long distance.

**Figure 4 biomolecules-12-00205-f004:**
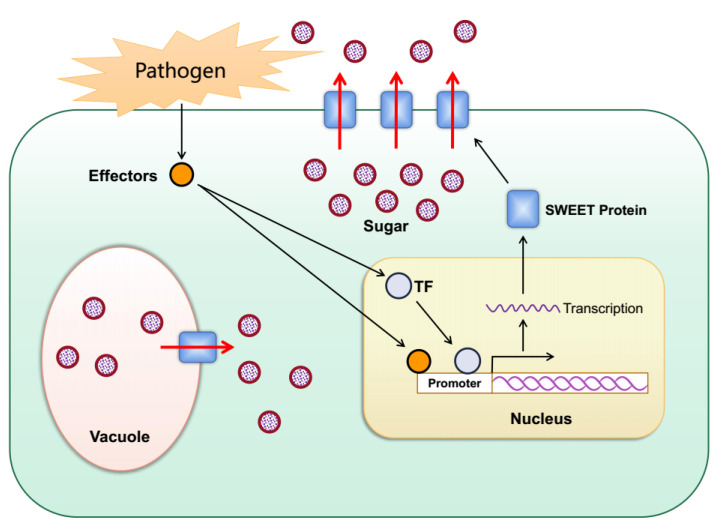
Schematic diagram for the role of plant SWEETs in pathogen nutrition. When bacterial or fungal pathogens invade plants, they secrete TAL effectors into host plant cells. The TAL effectors induce the expression of plant *SWEET**s* either directly or indirectly through activation of transcription factors, resulting in the outflow of sugar into the apoplast as a source of nutrition for the pathogens.

**Figure 5 biomolecules-12-00205-f005:**
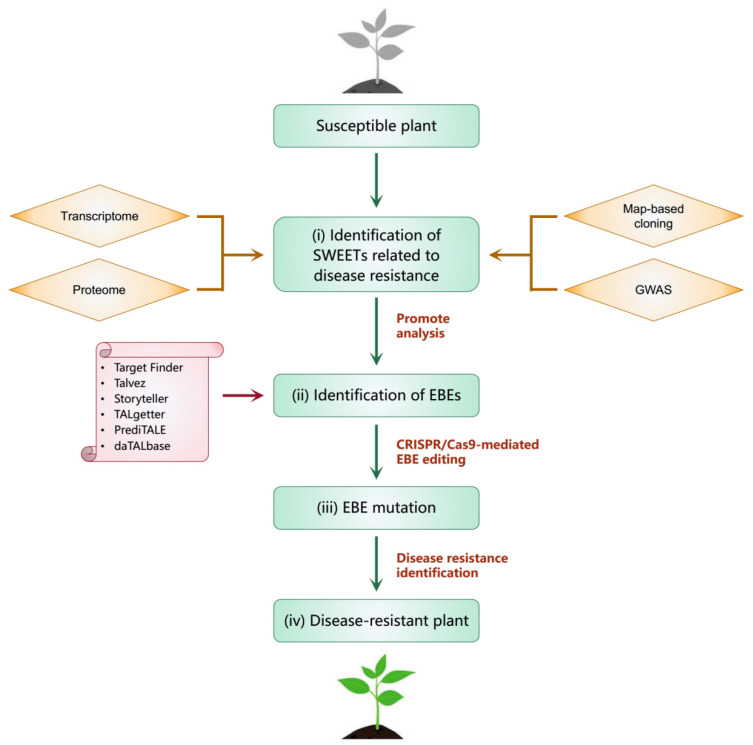
The strategy to obtain disease-resistant plants by modifying SWEET genes. It mainly includes the following steps (i–iv). (i) Identify SWEETs related to disease resistance through methods such as map-based cloning, genome-wide association analysis (GWAS), transcriptome and proteome. (ii) Identify and predict EBEs through tools such as PrediTALE, Storyteller, Talvez, TALgetter and Target Finder. (iii) Modify EBEs using gene-editing technology such as CRISPR-Cas9 system. (iv) Disease resistance test and obtain disease-resistant plants.

## Data Availability

Not applicable.

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
