# Peer review of "Plant SWEET Family of Sugar Transporters: Structure, Evolution and Biological Functions"

_biomolecules, 2022, doi:10.3390/biom12020205_

Round 1

Reviewer 1 Report

Paper with important matter, but the Importance of storage of sugars as a starch is not described clearly. The involvement of ADP and ATP in physiology of sugar is not clearly described. In the paper are mainly genetic aspect and not enough plant physiology regulations aspects. It must be better supported by the evolutionary aspects.

Line 10. Correct the style of “Sugars Will Eventually be Exported Transporters (SWEETs)”.

Lines 18-19. “…to help elucidate the molecular basis of their function in plants in the future.” It is no sense in function of plants in the future. Improve the style.

Line 318. Scientific name of plant must be in italic.

Line 328. “using genetic engineering may be a good way to increase the sugar content in tomato fruit...”. It is not a good way to use engineering for this less important property. In human nutrition other properties are in modern time more appreciated than just sugar concentration.  In many countries (especially in Europe, and in export to Europe) many customers (in Europe about 80%)  are no more appreciating food products obtained after being manipulated by genetic engineering.

Reviewer 2 Report

[Biomolecules] Manuscript ID: biomolecules-1538568
Lines 23-24, in the introduction, just in the first lines, is stated that “Sugar is an important energy source for
plants and a necessary carbon source for the synthesis of many of their metabolic intermediates [1-2]”.
However, the two references are not adequate at all, because they refer to sugar transporters and not to the
importance of sugar as an energy and carbon source in plants. Also, the first one, from 1998, refers to sugar
transporters in other organisms, but not plants (“Sugar transporters from bacteria, parasites and mammals:
structure-activity relationships”).
Lines 26- refer “… and plays an important role in signal transduction and resistance to stress [3-5]”, but the
references are not related with the role of sugars in resistance to stress. I think it should be more correct “…
and plays an important role in signal signaling [3-5]”. Also, more recent references could be included (e.g. Wang
M, Zang L, Jiao F, Perez-Garcia M-D, Ogé L, Hamama L, Le Gourrierec J, Sakr S and Chen J (2020) Sugar Signaling
and Post-transcriptional Regulation in Plants: An Overlooked or an Emerging Topic? Front. Plant Sci. 11:578096.
doi: 10.3389/fpls.2020.578096 )
Lines 27-35, the five references used to support the information given are from 1999 to 2002, and not the most
adequate. More recent and adequate information must be presented.
Lines 36-37, the references used must be more relevant and recent (for example doi: 10.1111/j.1365-
3040.2007.01708.x instead of ref.11, also from M. Stitt). The four references presented are from 1989, 1990,
1999, 2004.
Lines 40-41, stated that “Sucrose invertase metabolizes sucrose to monosaccharides [15].” However, sucrosesynthase
also catalyze the reversible cleavage of sucrose into fructose and UDP-glucose or ADP-glucose, having a
key role in sugar metabolism in plants (e.g. Stein O and Granot D (2019) An Overview of Sucrose Synthases in Plants.
Front. Plant Sci. 10:95. doi: 10.3389/fpls.2019.00095).
Lines 41-43, This phrase is not completely correct and should be more precise in the definition of source and
sink tissues. In fact, for example, young leaves under development can utilize light energy to photosynthesize
sugars and are sink tissues, because they need more sugars as a source of energy to support their growth
rate.
Lines 49-50, “…and SWEETs (Sugars will eventually be exported transporters)” should have a reference. I
suggest a review article.
Lines 54-55, the information given should be improved if the usual classification of transporters (symporter,
antiporter, cotransporters, uniporter) was used, including to SWEETs in the following lines.
Lines 56-57, “…new type of sugar transporter discovered in recent years that can transport sugars in two
directions and promote the diffusion of sugars along a concentration gradient [24,25]. Here is missing the
pioneer article from 2010 refereed as [28]. I´m not sure that a discover with more than a decade should be
consider “in recent years”, an affirmation also found in the abstract.
Lines 58-60, references are missing to support the sentence (“…regulating the transportation, distribution and
storage of carbohydrates.”) and I suggest at least these two recent reviews: Structure, evolution and diverse
physiological roles of SWEET sugar transporters in plants. doi: 10.1007/s11103-019-00872-4. Epub 2019 Apr
27.PMID: 31030374 Review; Structure and regulation of SWEET transporters in plants: An update. doi:
10.1016/j.plaphy.2020.08.043. Epub 2020 Aug 29.PMID: 32891967 Review.

Reviewer 3 Report

This manuscript by Ji et al. provides a review of the SWEET family of sugar transporters. The authors discussed the structural features, evolution, and the roles of SWEET transporters in plant development, stress tolerance, and other related processes. Overall, I think the manuscript is clearly written and well structured. I only have the following minor comments for the authors to consider in their revision:

Line 27 and throughout the text. “higher plants”. This should be avoided in modern biology. No plants are higher or lower. Please be specific.

Line 124. “replication fusion or horizontal gene transfer fusion”. What is replication fusion or horizontal gene transfer fusion? Please clarify.

Table 1c. How should the table be interpreted? More information should be provided in table description. For example, what do the different colors mean?

Line 174 and throughout the text. “branch”. What do you mean by “branch”? This is a very vague term. Are you referring to monophyletic groups or something else? Please be specific.

Figure 2. Values should be explained. Usually it does not make sense to include low support values since they are meaningless.

Lines 249-251. “The pollen development of plants with the OsSWEET11 gene that have been silenced remain in the mononuclear pollen stage or the dinuclear pollen stage;”. Don’t understand. Please consider revising.

Lines 275-276. “this gene may be a site related to …”. Don’t understand. Please clarify.

Round 2

Reviewer 1 Report

Lines 10-11.  Improve the text (style) to be correct.

Reviewer 2 Report

In the first round of revision for this paper it was collected enough review reports before I finished my revision because I was taking more time than expected. So, I only sent my revision of the introduction, the part than I saw that need more improvement. Now, in my opinion, the introduction still need an improvement. See  comments in the text uploaded, please. Also it must be stated that recently SWEETs in plants are the subject of other reviews, as for example, 
